# Mitsunobu Reaction: A Powerful Tool for the Synthesis of Natural Products: A Review

**DOI:** 10.3390/molecules27206953

**Published:** 2022-10-17

**Authors:** Saba Munawar, Ameer Fawad Zahoor, Shafaqat Ali, Sadia Javed, Muhammad Irfan, Ali Irfan, Katarzyna Kotwica-Mojzych, Mariusz Mojzych

**Affiliations:** 1Department of Chemistry, Government College University Faisalabad, Faisalabad 38000, Pakistan; 2College of Agriculture and Environmental Sciences, Government College University Faisalabad, Faisalabad 38000, Pakistan; 3Department of Biochemistry, Government College University Faisalabad, Faisalabad 38000, Pakistan; 4Department of Pharmaceutics, Government College University Faisalabad, Faisalabad 38000, Pakistan; 5Laboratory of Experimental Cytology, Medical University of Lublin, Radziwiłłowska 11, 20-080 Lublin, Poland; 6Department of Chemistry, Siedlce University of Natural Sciences and Humanities, 3-go Maja 54, 08-110 Siedlce, Poland

**Keywords:** natural products, Mitsunobu reaction, alkaloids, lignans, polyketides, amino acids, carbohydrates

## Abstract

The Mitsunobu reaction plays a vital part in organic chemistry due to its wide synthetic applications. It is considered as a significant reaction for the interconversion of one functional group (alcohol) to another (ester) in the presence of oxidizing agents (azodicarboxylates) and reducing agents (phosphines). It is a renowned stereoselective reaction which inverts the stereochemical configuration of end products. One of the most important applications of the Mitsunobu reaction is its role in the synthesis of natural products. This review article will focus on the contribution of the Mitsunobu reaction towards the total synthesis of natural products, highlighting their biological potential during recent years.

## 1. Introduction

The Mitsunobu reaction is considered as an important reaction because of its great applications in organic synthesis. It was discovered by Oyo Mitsunobu during the mid-twentieth century. It is a dehydrative redox reaction that converts an alcohol to an ester by coupling with an acid/pronucleophile [1]. The pronucleophiles include oxygen pronucleophiles, such as carboxylic acids, nitrogen pronucleophiles, such as imides and sulfonamides, and sulfur pronucleophiles, such as thiols, etc. [2]. This reaction is mediated by phosphines, i.e., triphenylphosphine, and azodicarboxylates, i.e., diethyl azodicarboxylate (DEAD), diisopropyl azodicarboxylate (DIAD), etc., and a solvent [3]. A general Mitsunobu reaction is represented in Figure 1.

In case of sterically hindered alcohols, 4-nitrobenzoic acids or phthalimides are also used for coupling. Besides these, many other reagents, such as fluorinated alcohols and hydroxamates, can also be employed for Mitsunobu coupling [4]. In this way, this reaction allows the formation of C-C, C-N, C-O and C-S bonds formation owing to its diverse selection of reagents. The panoramic feature of this reaction is that it brings about inversion of configuration by coupling of a chiral secondary alcohol and a pronucleophile [5].

The Mitsunobu reaction has a wide range of applications in organic synthesis as it encapsulates the total synthesis of natural products, drugs, analogues, and semisynthetic derivatives of naturally occurring compounds, etc. It constitutes a key step in the synthesis of many of the natural products and other pharmaceuticals. Figure 1 describes a few examples where the Mitsunobu reaction has played its crucial role:

The total synthesis of sinodielide A **1**, which is used as a flavoring, fragrance, medicine, and poison, employs the Mitsunobu reaction as an important step [6,7]. Similarly, the synthesis of anti-proliferative menaquinone **4**, which is a homologue of vitamin K, and synthesis of cannabidiol derivatives **3**, which is used for the treatment of arthritis, immune system disorders, and diabetes, include the Mitsunobu reaction as a key step [8,9]. It has played a great role in the synthesis of drugs and their derivatives, such as the synthesis of NPS R-568 **2**, which is a calcimimetic drug used for the treatment of primary and secondary hyperparathyroidism [10]. This paper reviews the importance of the Mitsunobu reaction in the synthesis of some of the natural products reported in recent years.

## 2. Review of the Literature

### 2.1. Alkaloids-Based Natural Product Synthesis

#### 2.1.1. Yuzurimine Alkaloids

The Mitsunobu reaction has been used for the modification and synthesis of scaffolds for the total synthesis of natural products. The yuzurimine alkaloids are extracted from *Daphniphyllum macropodum* [11] and exhibit many biological activities. In 2019, Hayakawa et al. proposed the synthesis of the heterocyclic portion of yuzurimine-type alkaloids [12]. For this purpose, a protected alcohol **6** synthesized from precursor **5** underwent the Mitsunobu reaction for intramolecular alkylation and gave Ns amide **7** in a 69% yield. The next step involved selective deprotection of hydroxyl group to give **8** in a 75% yield. An intermolecular Mitsunobu reaction was performed to give a bicyclic product **9**, which could be converted to the desired product deoxyyuzurimine **10** over a few steps (Figure 2).

#### 2.1.2. Phenanthrene Alkaloids

Phenanthrenes are known as chromophores that exhibit many physiochemical properties, such as photoconductance and electroluminescence [13]. Aristolactams are phenanthrene-related compounds that are isolated from *Aristolochia argentina*. These have been used in many folk medicines in the past [14]. Luong et al. reported the total synthesis of aristolactam **15** in 2019 [15]. To achieve the task an appropriate aldehyde **11** was converted to phenol **12** over a few steps. In the next step, an already synthesized alcohol **13** with an *anti*:*syn* ratio of 6:1 was utilized to couple with phenol **12** via a Mitsunobu reaction in the presence of diisopropyl azodicarboxylate (DIAD) and triphenyl phosphine (PPh_3_) in toluene at 0 °C to 90 °C to obtain 76% of the coupled product **14**. The last step included the global deprotection and cyclization of ether by using TFA in anisole that provided stereoselective (+)-aristolactam **15** in a 32% yield (Figure 3).

#### 2.1.3. Indole Alkaloids

Nauclefine is an indole alkaloid which was extracted from a West African tree named *Nauclea latifola*. It exhibits a number of biological activities, such as anti-tuberculotic, anti-inflammatory, and anti-neurodegenerative [16,17]. In 2021, Chen et al. proposed the total synthesis of nauclefine **24** by using commercially available starting materials [18]. The total synthesis was accomplished by employing the Mitsunobu reaction, Fischer indolization, and allylic oxidation in six steps. In the first step, the TMS-protected alkynol **16** was made to react with the hydroxy isoindolene derivative **17** via a Mitsunobu reaction in the presence of PPh_3_ and DIAD in THF at 0 °C to room temperature to give alkyne **18** in a 94% yield. In the next step, the alkyne **18** was reacted with hydrazine monohydrate using MeOH/DCM and freshly prepared nicotinic acid in the presence of K_2_CO_3_, EtOAc/H_2_O to yield nicotine amide followed by catalysis in the presence of 2.5 mol% of (Cp*RhCl_2_)_2_ as a catalyst and 3 eq. of CsOAc with TFE to furnish regioisomeric products **20** and **21** in the yield of 28 and 69%, respectively. Later on, naphthyridine derivatives were treated via Mitsunobu reaction conditions with PPh_3_, DIAD, and THF in a reaction mixture to give tricyclic product **23** in a 76% yield. It was further oxidized in the presence of SeO_2_ and Dess–Martin periodinane followed by Fischer indolization to yield the desired product, nauclefine **24** in a 66% yield (Figure 4). 

#### 2.1.4. Discorhabdin Alkaloids

Discorhabdin is an alkaloid extracted from marine sponges and possesses anti-cancerous, anti-malarial, and antimicrobial properties, etc. [19]. Noro et al. explained the total synthesis of discorhabdin by employing a number of synthetic strategies in 2021 [20]. The main framework of discorhabdin consists of an aza-spirodienone-fused pyrroloiminoquinone having a pentacyclic core. In the first step, commercially available carboxylic acid **25** was reduced with LiAlH_4_ in THF followed by Birch reduction using Li and NH_3_ in EtOH/THF to yield cyclohexadiene **26**. The acidic hydrolysis of cyclohexadiene pursued by ketone reduction by NaBH_4_ and the selective protection of primary alcohol using TBSCl in DCM to yield cyclohexenol **27** in a 70% yield. Then, tryptamine derivative **28** was coupled with cyclohexenol **27** via the Mitsunobu reaction in the presence of DIAD and PPh_3_ in toluene to yield **29**. Further, the compound **29** was deprotected to yield **30** in quantitative yield that again was subjected to another Mitsunobu reaction in the presence of NsNHBoc, DIAD, and PPh_3_ to yield Boc-protected tryptamine derivative **31** in a 67% yield. The tryptamine derivative **31** was then converted to discorhabdin **32** over a few steps (Figure 5).

#### 2.1.5. Lycopodium Alkaloids

Fawcettimine belongs to the Lycopodium alkaloid and is extracted from *Lycopodium fawcetti* [21]. This class of compounds consists of approximately 300 natural products, some of which are biologically active, e.g., fawcettimine. In 2020, Zeng et al., reported the total synthesis of fawcettimine over eleven steps using commercially available starting material (*R*)-(+)-pulegone [22]. For achieving the total synthesis of fawcettimine, Pd-mediated cycloalkenylation, Mitsunobu reaction, and an oxa-Diels–Alder reaction were utilized as key steps. (*R*)-5-Methyl-2-cyclohexen-one **33** was converted to intermediate **34** after few steps. It was further allowed to undergo the oxa-Diels–Alder reaction using EuFOD in the first step and para toluene sulphonic acid (PTSA) in acetone in the next step to obtain the aldehyde derivative **35** in a 93% yield. The aldehyde derivative **35** was further reduced with NaBH_4_ in DCM/MeOH and subjected with MsCl in Et_3_N and DCM to obtain mesylate **36** in a 64% yield. In the following step, mesylate **36** was aminated using *p*-NH_2_Ns in K_2_CO_3_. Subsequently, Dess–Martin oxidation furnished the desired intermediate **37** in a 72% yield. The intermediate **37** underwent the Mitsunobu reaction for ring closure by using PPh_3_, DEAD. and PhMe to obtain a 67% yield of **38**. In the last step, desulfonylation of **38** was performed by using PhSH in MeCN and KOH to obtain the final product, fawcettimine **39**, in an 81% yield (Figure 6).

Lycopoclavamine-A [23] resembles fawcettimine-type alkaloids [21] having minor structural differences. Kaneko et al. reported the total synthesis of lycopoclavamine-A in 2019 with overall yield of 14.4% by featuring a stereoselective conjugate addition and the Pauson–Khand reaction as the main steps [24]. The total synthesis was initiated from crotonamide **40**, which was converted to diastereomer **41** over a few steps. The Mitsunobu reaction of diastereomer **41** was performed in the presence of diethyl azodicarboxylate (DEAD) and triphenyl phosphine to introduce nitrogen in the ring to yield bicyclic compound **42** in a 92% yield. The next step involved the deprotection of group OTBDPS and the generation of the primary alcohol, which underwent an intramolecular Mitsunobu reaction to yield the tricyclic product **43** in a 74% yield. In the following step, two MOM groups were removed and a triflate group was deprotected in the presence of 12 N HCl and 1 N NaOH with dioxane in methanol to yield secondary alcohol **44**. In the last step, the nosyl group was deprotected to yield the final product **45** in a 79% yield (Figure 6). An epimer of **41** 3-*epi*-24 underwent a similar strategy to provide the final product, lycopoclavamine-A, contributing to the overall yield (Figure 7). 

Palhinine alkaloids belong to Lycopodium alkaloids [21], having a fused isotwistane and azonane in their skeleton. A number of strategies have been developed for the total synthesis of a palhinine skeleton. Han et al. reported the synthesis of a tetracyclic skeleton of palhinine alkaloids by employing the carbocyclization cascade mechanism in 2020 [25]. The first step involved the reaction of 1,3-cyclohexanedione **46** with 1,3-dibromopropane **47** followed by α-propargylation in the presence of 3-bromo-trimethylsilyl propyne **48** to yield **49** an 83% yield. In the next step, 1,2-addition of enone took place preceded by acid hydrolysis that provided 64% of compound **51**. Further, an intermolecular Mitsunobu reaction in the presence of PPh_3_, DIAD, and NsNH_2_ in THF followed by desilylation produced **52** a 74% yield. Another intramolecular Mitsunobu reaction in the presence of triphenyl phosphine and diethyl azodicarboxylate was performed to afford **53** a 70% yield. The last step included the carbocyclization cascade to produce a diastereomeric tetracyclic skeleton **54** in a 28% yield and a by-product **55** in a 24% yield. The tetracyclic skeleton **54** could serve as a key intermediate for the total synthesis of palhinine alkaloids (Figure 8).

#### 2.1.6. Pyrrole Alkaloids

Parvistemonine is an alkaloid extracted from the roots of *Stemona parviflora* and has been used as a folk medicine. Matsuo et al., in 2020, reported the total synthesis of parvistemonine in ten steps by employing the Mukaiyama–Michael addition, an aza-Wittig reaction, a Paal–Knorr pyrrole synthesis and the Mitsunobu reaction, etc., to give the final product in a 19.6% overall yield [26]. The first step involved the reaction of γ-lactone **56** with n-butyllithium and then oxidation with IBX to produce keto aldehyde **57** in a 74% yield. It was further treated with azide **58** in the presence of triphenyl phosphine to give the required coupling product **59** in a 65% yield. The next step included acid-mediated annulation in the presence of toluene sulphonic acid and acetonitrile followed by deprotonation with LHMDS and methylation with MeI, which gave 83% of the product **60**. Later on, the compound **60** was subjected to alkaline hydrolysis and then was reacted with di-2-methoxyethyl azodicarboxylate (DMEAD) and PPh_3_ to give the final product **61** in a 77% yield (Figure 9).

#### 2.1.7. Oroidin Alkaloids

Nagelamides belong to a group of oroidin alkaloids [27] and are derived from a marine sponge. Bhandari et al., in 2020, proposed the total synthesis of nagelamides by employing the Mukaiyama–Michael addition, an acid-mediated annulation, an aza-Wittig reaction, and a Mitsunobu reaction as the key steps [28]. First of all, two fragments of imidazolyl iodide **63** and vinyl stannanes **65** were synthesized from diiodo imidazole **62** and imidazole iodide **64** in a 93% and 57% yield, respectively. These two fragments, **63** and **65,** were coupled together in the presence of Pd_2_dba_3_ CsF, CuI, PPh_3_, and DMF to give 75% of the coupled product **66**. Further, **66** was treated with an excess of n-butyllithium and then an azido group was introduced into both imidazole groups followed by desilylation by using TBAF and THF to obtain diol **67** in a 78% yield. The Mitsunobu reaction was used to couple diol **67** and the dibromo dione derivative **68** in the presence of PPh_3_, DIAD, and THF to obtain the dibrominated pyrrole derivative **69** in an 80% yield. The pyrrole derivative **69** was hydrolyzed by NaOH and deprotected by acidic methanol followed by the hydrogenolysis of azides to amino groups by using a Lindlar catalyst to produce nagelamide D **70** in a 51% yield over two steps (Figure 10).

#### 2.1.8. Communesin Alkaloids

Communesin alkaloids are polycyclic epoxide-containing alkaloids which exhibit a number of biological activities. These alkaloids were first extracted from *Penicillium* fungus. These alkaloids are cytotoxic, insecticidal, and anti-cancerous in their behavior [29]. The total synthesis of communesin alkaloids have been a challenging task for researchers due to their complex epoxide-containing structures. In 2019, Pompeo et al. proposed the total synthesis of two natural and six synthetic communesin alkaloids [30]. The total synthesis commenced from sulfonamide **71**, which was condensed with *N*-methyl-4-bromoistan **71**, followed by allylation with allyl magnesium bromide **72** along with ozonolysis in the presence of ozone and methanol to give **73** in an 85% yield. Further, the Mitsunobu reaction with *N*-carbobenzoxy-2-nitrobenzenesulfonamide gave 76% of the product **74**. Over a few steps, **74** was converted to heterodimeric diamine **75**. Finally, heterodimeric diamine **75** was reacted with tertiary butoxy lithium and neutralized with pyridinium *p*-toluene sulphonate (PPTS) along with TASF treatment to give communesin A **76** in a 77% yield. Similarly, heterodimeric diamine **75** underwent aminal rearrangement followed by acylation with sorbic anhydride to furnish communesin B **77** in an 86% yield (Figure 11). The synthesized communesin alkaloids were selected for anticancer activities against five human cancer cell lines, namely, prostate carcinoma (DU-145), human lung carcinoma (A549), colorectal carcinoma (HCT-116), breast adenocarcinoma (MCF7), and cervical adenocarcinoma (HeLa) cell lines. Among the synthesized communesin alkaloids, communesin alkaloid B manifested the best cytotoxic potency, having IC_50_ values as presented in Figure 2. 

#### 2.1.9. Pyrroloquinoline Alkaloids

Marine alkaloids, such as batzellines and isobatzellines [31], are renowned for their various biological activities. Deep water marine Caribbean sponge *Batzella* sp. Acts as a source of batzellines, which possess cytotoxic and anti-HIV activities. Yamashita et al. reported the total synthesis of batzellines in 2020 by employing a number of key steps, such as ring expansion of benzocyclobutenone oxime sulfonate and benzyne-mediated cyclization to design an efficient route [32]. The total synthesis began with the reaction of ketene silyl acetal **79** and 4-bromoveratrole **78** in the presence of LiTMP and THF, followed by subsequent treatment with aq. HF to give the primary alcohol **80** in an 84% yield. The next step involved the amide formation via Mitsunobu reaction conditions with NSNHBoc and DMEAD in the reaction mixture to obtain 78% amide derivative **81**. Further, bromination of amide **81** with an NBS agent followed by condensation in the presence of NH_2_OH.HCl and Boc protection with acetonitrile in DMAP gave oxime sulphonate **82**. It was treated with NaSMe in MeCN preceded by cascade cyclization using an excess of LiTMP to pivotal aryl anion **83**. The pivotal aryl anion **83** underwent an aqueous work up to give pyrroloindoles **85** and **86** in an 80 and 82% yield, respectively. Isobatzelline B **87** was synthesized from pyrroloindoles **85** by treatment with TFA followed by amino group addition to provide a 94% yield. The compound **86** was treated with a manganese reagent in the presence of oxygen to give batzelline A **88** along with a fragment **89,** which, upon subsequent treatment with ammonium chloride and methanol, yielded isobatzelline A **90** in a 74% yield (Figure 12).

#### 2.1.10. Morphine Alkaloids

Morphine is an analgesic alkaloid isolated from the opium poppy plant. It has been ranked among the top medicines by the World Health Organization. Pharmacologically, it acts as an analgesic and sedative medicine [33]. The total synthesis of morphine, comprising of 16 steps, was first proposed by Zhang et al. in 2019 [34]. It was commenced from 3-butyn-1-ol **91**, which was undergone for primary hydroxyl group protection, followed by the formation of a Weinreb amide **92** upon reaction with *N*-methoxy-*N*-methyl carbamoyl chloride. In the next step, boronation of the Weinreb amide **92** along with Suzuki coupling, followed by the addition of methyl magnesium bromide, gave 67% of the enone precursor **94**. After the screening of a number of catalysts, 2,4,6-triisopropyl benzoic acid was used as the most effective catalyst along with an additive **95** for an enantioselective Michael addition followed by Robbinson annulation to give the tricyclic product **96**. In the later step, a tetracyclic enone **98** was formed after the allylic group introduction along with Friedal-Crafts type cyclization. Later on, selective epoxidation followed by treatment with N_2_H_4_.H_2_O and debenzylation in the presence of DDQ gave 52% of the desired alcohol **99**. The alcohol **99** was converted to codeine **100** in a 68% yield by highly a regioselective intermolecular Mitsunobu reaction followed by redox reactions. In the last step, the demethylation of codeine in the presence of boron tribromide gave the desired morphine **101** in an 81% yield (Figure 13).

Thebainone A is related to the family of morphine alkaloids [35]. It is extracted from *Papaver sominiferum*. Thebainone A has many applications in the medicinal field. It is used to relieve severe pain. Wang et al., in 2020, reported the total synthesis of (±)-Thebainone A [36]. The total synthesis commenced from isovaniline **102**, which was converted to isoxazolidine **103** over a few steps. Next, isoxazolidine **103** was hydrogenolytically cleaved to obtain trihydroxy phenol **104**. The trihydroxy phenol **104** was made to undergo the Mitsunobu reaction with diethyl azodicarboxylate (DEAD) and Et_3_N.HCl to obtain morphinan **105** in a 52% yield. The *cis*-diol moiety in **105** underwent formamide acetal pyrolysis followed by esterification and then conversion to ethyl carbamate **106** in an 89% yield. The ethyl carbamate **106** was made to react with selenium dioxide and oxidized with Dess–Martin periodinane to get the enone **107** in a 21% yield. In the next step, the enone **107** was protected as dioxolane followed by reduction and deprotection to yield the final product (±)-Thebainone A **109** in a 96% yield (Figure 14). 

(−)-Thebainone A has also been synthesized by Hou et al. in 2021 [37]. They used a deconstructive methodology to achieve the total synthesis of (−)-thebainone A with enantioselective ratio of up to 99.5:0.5 by employing a C-O bond cleavage reaction and enantioselective C-C bond activation as key steps. The total synthesis commenced from a known anisole **110**. It was converted into an alcohol derivative **111** through Birch reduction and formation of ketal in the presence of pyridinium *p*-toluene sulfonate (PPTS). In the next step, the Mitsunobu coupling with an alcohol **112** provided the derivative of the ketal substrate **113** in a 93% yield. In the next step, rhodium-catalyzed coupling with (*R*)-DTBM-segphos produced a tetra cycle **114** in a 76% yield, which, after a few steps, gave (−)-thebainone **115** (Figure 15). 

#### 2.1.11. Calciphylline Alkaloids

Daphniphyllum is a large group of alkaloidal natural products. Calyciphylline-type alkaloids belong to this class of compounds. These are obtained from the leaves of *Daphniphyllum calycinum* and stems of *D. subverticillatum* [38]. These are biologically active substances and exhibit activity as cytotoxic substances against murine lymphoma (L1210) cells, and inhibit the aggregation of platelets. The synthesis of a common calyciphylline A, E bicyclic core was proposed by Kumar et al. in 2019 [39]. The key steps of this synthesis include the Ireland–Claisen rearrangement, the Mitsunobu reaction, an intramolecular aldol, and an aza-Michael reaction. The synthesis was started from cyclohexenol **116**, which had undergone esterification with propanoyl chloride, followed by the Ireland–Claisen reaction and bromo lactonization to furnish lactone **117** and **118**. The lactone **117** was converted to lactone **118** by isomerization and quenching with diethyl malonate. It was further undergone for reductive cleavage with Zn/EtOH to give alcohol **119** in a 70% yield. In the next step, a Mitsunobu conversion was performed in the presence of NsNH_2_ with PPh_3_ and DIAD at room temperature for 24 h to give sulfonamide **120** in a 65% yield. In the later step, oxidative cleavage via Jin’s protocol was employed, followed by treatment with piperidinium acetate to give aldehyde **122** in a 45% yield. In the last step, an intramolecular aldol reaction with sodium chlorate, along with a subsequent aza-Michael reaction and Pinnick oxidation in the presence of allyl bromide and trimethyl amine, furnished the bicyclic compound **123** in an 80% yield (Figure 16). 

#### 2.1.12. Aspidosperma Alkaloids

The Mitsunobu reaction constitutes an important step in the total synthesis of natural products. Aspidosperma-derived monoterpene alkaloids are complex and biologically active natural products [40]. Liu et al. proposed the total synthesis of these monoterpene alkaloids in 2019 [41]. The total synthesis of mersicarpene was initiated from 2-methylaniline **124**, which was converted to substituted indole (−)-**125** over a few steps. The synthesized indole (−)-**125** underwent the Mitsunobu reaction in the presence of 1.02 equivalents of diphenylphosphonic azide, 1.2 equivalents of diisopropyl azodicarboxylate, and 1.2 equivalents of triphenyl phosphine at 0 °C in THF to replace the OH group with an azide group, and afford azidoindole (+)-**126** in an 98% yield. The next step involved the hydroboration of C=C in azidoindole (+)-**126** to obtain the compound (−)-**127** in a 71% yield. In the later step, 68% of *N*-acyl indole (+)-**128** was obtained via Ley oxidation in the presence of NMO, TPAP, and methyl cyanide. The oxidation of the *N*-acyl indole via Kerr’s method followed by Staudinger–aza-Wittig reaction yielded 64% of (−)-mersicarpene **129** (Figure 17). 

Alkaloid-based natural products possess potent medicinal properties, and thus are employed for the treatment of a number of disorders. Alkaloids act as anti-bacterial, anti-fungal, and anti-cancerous agents, and against various other diseases, etc. *Aspidosperma* and *Kopsia* alkaloids are biologically active and structurally diverse compounds [42]. The total synthesis of kopsifoline D, beninine, and deoxoapodine was reported by Zhou et al. in 2019 [43]. The domino deprotection-Michael addition-nucleophilic substitution, Johnsan–Claisen rearrangement, and Corey–Bakshi–Shibata reduction reactions were employed as key steps to achieve the core framework of the aforementioned alkaloids. The total synthesis of (−)-kopsifoline D, (−)-beninine, and (−)-deoxoapodine was commenced from cyclohexan-1,3-dione **130**, which was converted to allylic alcohol **131** over a few steps. The next step involved the Mitsunobu reaction with 1.1 equivalents of (2-chloroethyl)-2,4-dinitrobenzenesulphonamide **132** in the presence of 1.1 equivalents of diisopropyl azodicarboxylate (DIAD) and 1.05 equivalents of triphenyl phosphine (PPh_3_) to give sulfonamide **133** in a 90% yield. The sulfonamide **133** further underwent allylic oxidation in the presence of chromium trioxide and *tert*-butyl hydroperoxide followed by intramolecular nucleophilic substitution to give tricyclic aminoketone **134** in a 70% yield. In the later step, 76% of indoline **135** was obtained by treating tricyclic aminoketone with phenylhydrazine followed by reduction with LiAlH_4_. Further, Swern oxidation of indoline **135,** accompanied by deprotection with TBAF, gave alcohol **136** in an 82% yield. The last step involved oxymercuration and demercuration in the presence of Hg (OTFAc)_2_ to give 75% of deoxoapodine **137,** and transannular cyclization in the presence of *tert*-butoxide and TsCl to obtain kopsifoline D **138** in a 46% yield. The synthesis of (−)-beninine was commenced from tricyclic aminoketone **134**, which was reacted with o-methoxyphenyl hydrazine followed by reduction with triethyl silane and trifluoroacetic acid along with reduction by LiAlH_4_ and direct acetylation to furnish acetate **139** in a 61% yield. Finally, acetate **139** underwent selective hydrolysis followed by oxymercuration and demercuration in the presence of Hg (OTFAc)_2_ accompanying the removal of the acetyl group with hydrochloric acid to provide (−)-beninine **140** in an 87% yield (Figure 18). 

Leuconodines are aspidosperma-derived natural products [40]. These belong to the family of monoterpene alkaloids possessing a rare fenestrene core in their structure. These are biologically active compounds as these exhibit moderate cytotoxic properties and have found usage in traditional medicine for the treatment of yaws bacterial infections and worm infections. Leuconodine E manifests an important role as a cytotoxic substance possessing an IC_50_ value of 9.34 µg/mL when employed with 0.1 µg/mL vincristine [40]. Leuconodines have complex structures that pose many hurdles in their total synthesis. After finding out alternative methods to cope with synthesis problems, Zhang et al. proposed the total synthesis of leuconodine D and E in 2019 [44]. The total synthesis was commenced from commercially available tryptophol **141**, which was made to undergo TBS protection and was reacted with acyl chloride, followed by TBS deprotection and oxidative Heck cross coupling reaction in the presence of palladium acetate and ligands **143** and **144,** to give **145** in an 88% yield. Next, the Mitsunobu reaction was performed in the presence of 1.5 eq. of sulfonyl carbamate **146**, 1.5 eq. of diethyl azodicarboxylate (DEAD), and 1.5 eq. of triphenyl phosphine at 0 °C to rt for 10 h to give **147** in quantitative yield. In the next step, deprotection of the ortho-nitrosulfonyl group, followed by epoxidation in the presence of oxone and cyclization in acidic conditions, gave a mixture of diastereomer **148** and **149** in 1:1. In the next step, NaCN-mediated heating was used to cleave the methoxycarbonyl group and *N*-allylation with allyl iodide **150**, followed by an HG-II mediated ring closing metathesis and Pd-catalyzed hydrogenation, which gave leuconodine E **151** in an 88% yield. The hydroxy group of **151** was converted to xanthate accompanied by a Barton–McCombie reduction to furnish 66% of leuconodine D **152** (Figure 19). 

#### 2.1.13. Amaryllidaceae Alkaloids

(+)-Lycoricidine and conduramines belong to family of *Amaryllidaceae* alkaloids [45]. These alkaloids have gained much attention because of their biological significance. On account of their biological properties, these act as antitumor, antimicrobial, anti-fungal, and analgesic compounds [46]. Lo et al. proposed the total synthesis of lycoricidine and conduramines in 2019 by employing a diastereoselective mechanism [47]. The total synthesis of *ent*-conduramine [48] was commenced from cyclic diol **153**, which underwent [3,3] sigmatropic rearrangement for the azido group introduction to obtain allylic azide **154** in a 70% yield. The next step was used to obtain the required configuration via the Mitsunobu reaction in the presence of triphenyl phosphine and diethyl azodicarboxylate and benzyl ether to obtain a 1,4-*syn* type azido alcohol **155** in an 87% yield. The last step involved the reduction of the azide group with lithium aluminum hydride (LAH) accompanied by deprotection and peracylation to give 89% of tetraacetyl *ent*-conduramine F-1 **156** (Figure 20). 

### 2.2. Neolignans-Based Natural Product Synthesis

Plant-based lignans exhibit a number of biological activities, such as antileishmanial, antiparasitic, trypanocidal, and antitumor [49,50]. Surinamensinols are lignan-based natural products, which exhibit anti-inflammatory and anti-cancerous activities. Surinamensinol A and B demonstrate appreciable anti-cancerous activity against A549, SK-MEL-2, HCT-15, and SK-OV-3 cell lines [51]. Various research groups have reported the total synthesis of surinamensinols consisting of lengthy schematic pathways. In 2020, Avula et al. reported an expeditious method for the total synthesis of surinamensinols by using commercially available starting materials consisting of six steps [52]. The first step involved the reaction of allyl alcohol **157** and benzyl bromide in NaH and anhydrous THF to give a 98% product, which was further reacted with 9-borabicyclo [3.3.1] nonane solution (9-BBN) in THF, followed by reaction with 4-bromo-2-methoxyphenol **158** in Pd(PPh_3_) and THF via Suzuki–Miyaura coupling to yield 86% of phenol benzyl ether **159**. In the following step, phenyl benzyl ether **159** was made to couple with (*S*)-ethyl lactate **160** by Mitsunobu reaction in the presence of diisopropyl azodicarboxylate (DIAD), triphenyl phosphine, and anhydrous THF under reflux for 24 h to give the required chiral ester **161** in the yield of 88%. The compound **161** was then reduced by using DIBAL-H, and then subjected to a reaction with 3,4,5-trimethoxyphenyl magnesium bromide to yield a *anti* and *syn* diastereomeric mixture of products **162**. This mixture of products was then deprotected by using 10%Pd/C and ethyl acetate. The final products **163** and **164** were obtained in overall 14% and 22% yields, respectively (Figure 21). 

Lignans are natural compounds which have a number of medicinal properties, i.e., anticancer, antimicrobial, anti-inflammatory, antiviral, antifungal, and neurotoxic etc. [53]. Aglacins are referred to as aryl tetracyclic lactone lignans, which are isolated from *Aglaia cordata*. Xu et al. proposed the total synthesis of aglacins by employing an asymmetric photoenolization/Diels–Alder (APEDA) reaction as a key step, in 2021 [54]. 2-Methyl benzaldehyde **165** and bromo furan derivative **166** were combined to give compound **167** over a few steps, which was reduced in the presence of LiAlH_4_ to yield **168 in** an 88% yield. A Mitsunobu reaction was employed in the presence of triphenyl phosphine and *di*-*tert*-butyl-azodicarboxylate to form a cyclic ether ring followed by TIPS deprotection by using tetrabutylammonium fluoride to obtain (−)-aglacin E **169** in a 50% yield. Aglacin E was further converted into (−)-aglacin A **170** in a 49% yield by employing the Mitsunobu reaction and (−)-aglacin B **171** in a 90% yield by reductive dihydroxylation (Figure 22). 

Furthermore, synthesis of (+)-linoxepin **177** commenced from the APEDA reaction of **172** and dienophile **166**, which provided 76% of tricyclic product **173** with 93% enantioselectivity. Next, the desired triflate **174** was obtained by oxidation in the presence of DMP and sodium bicarbonate accompanied by triflation in the presence of Tf_2_O in an 84% yield over two steps. In the following step, a Pd-catalyzed Suzuki–Miyaura coupling reaction between triflate **174** and compound **175** was performed in the presence of Pd(PPh_3_), 1,4-dioxane and phosphoric acid to obtain the desired coupling product **176** in a 60% yield. The next step involved the removal of TBS and two methyl groups followed by an intramolecular Mitsunobu reaction for seven-membered cyclo-ether ring formation along with the addition of a methyl group in the presence of methyl iodide to obtain the final product (+)-linoxepin **177** in a 40% yield over two steps (Figure 23). 

Lignans are an important class of natural products that possess a number of biological activities, such as antitumor, anti-inflammatory, anti-bacterial, and immunomodulators. 7*S*-HMR is converted to (−)-enterolactone by intestinal bacteria and exhibits chemo preventive effects on the development of mammary carcinoma induced by DMBA. Furthermore, it possesses anti-oxidant potential and exhibits chemopreventive effects in an ApcMin mice model of human familial adenomatous polyposis [55]. These are present in many fruits, vegetables, and seeds. 7*R*-Hydroxymatairesinol belongs to lignans which have polyphenolic structures. It is reported to have chemotherapeutic and metabolizing effects. In 2021, Colombo et al. reported the preparation of a 7*R*-HMR isomer from a 7*S*-HMR isomer using kinetic reduction, Mitsunobu reaction for epimerization, and ester hydrolysis as key steps [56]. In the first step, the protection of phenolic functional groups in the 7*S*-HMR **178** isomer was completed by using TBSCl and imidazole in DMF to obtain alcohol **179** in a 68% yield. In the next step, a Mitsunobu reaction was performed in the presence of PPh_3_, DIAD, THF, and *p*-nitrobenzoic acid at room temperature for 24 h to obtain **180** with an inverted OH group in a 43% yield. The final step involved hydrolysis to ensure complete inversion, followed by deprotection using TBAF and AcOH in THF to obtain 7*R*-HMR **181** in a 72% yield (Figure 24). 

Neoflavones are naturally occurring compounds and act as important scaffolds for the synthesis of drugs and biologically active compounds. Prenyl groups enhance the bioactivity of respective compounds. They exhibit protease inhibition in COVID-19 [57]. Prenylneoflavones are derivatives of neoflavones whose synthesis has been reported by Lozinski et al. in 2020. They employed a Mitsunobu reaction, olefin cross-metathesis reaction, and Claisen rearrangement as key steps to obtain a 5% overall yield in six steps [58]. The first step involved the conversion of 3-methoxyacetophenone **182** to neoflavone **183** in a 52% yield via Claisen rearrangement in the presence of diethyl carbonate, and Pechmann condensation in the presence of resorcinol in sulfuric acid. Next, the Mitsunobu reaction with allyl alcohol in the presence of diazo-dicarboxylate (DIAD) and triphenyl phosphine at 0 °C gave 7-allyloxy-neoflavone **184** in a 78% yield. Further, Claisen rearrangement with Eu(fod)_3_ along with acylation gave an acetylated product **185** in a 79% yield. The 7-acetoxyneoflavone was further treated with Grubbs second generation catalyst to yield prenylneoflavone **186** in a 65% yield (Figure 25). 

Lignans are polyphenolic compounds that possess many biological and medicinal properties [59]. Ligraminol D and E are lignans which are isolated from *Acorus. gramineus* and exhibit anti-proliferative activity against tumor cells and inhibit the production of nitric oxide in BV-2 cells [60]. Mane et al. proposed the total synthesis of Ligraminol D and E in 2019 [61]. The synthesis of Ligraminol E was commenced from known aldehyde **187**, which was converted to a chiral diol **188** over a few steps. Next, diol **188** was made to react with benzoyl chloride for the selective protection of the hydroxyl group and production of secondary alcohol **189** in a 90% yield. It was further treated with acetate **190** via Mitsunobu reaction in the presence of DEAD and PPh_3_ in anhydrous THF under reflux for 6 h to give 88% of ether **191**, which was hydrogenated in the presence of Pd/C followed by a global reduction to yield Ligraminol E **192** in a 93% yield (Figure 26). 

For the synthesis of Ligraminol D, alcohol **194** obtained from aldehyde **193** over a few steps was made to undergo a Mitsunobu reaction with ester **190** in the presence of PPh_3_ and DEAD in anhydrous THF under reflux for 6 h to give **195** in an 87% yield. It was converted to Ligraminol D **196** by debenzylation in a 93% yield (Figure 27).

#### Sesquineolignans

Princepin and isoprincepin are sesquineolignans obtained from *Joanessia princeps*. These compounds show biological activity as anti-estrogenic and anti-oxidant compounds [62]. Kobayashi et al. explained the total synthesis of princepin and isoprincepin in 2019 [63]. The total synthesis was commenced from aldehyde **197** and alcohol **198**, which underwent condensation via a Mitsunobu reaction in the presence of DIAD and PPh_3_ at room temperature for 2 h, followed by pinacol conversion to give **199**. Next, the compound **199** underwent TIPS and benzyl ether deprotection followed by Dess–Martin oxidation to give isomer **200** in a 93% yield. A total of 85% of isomer **201** was obtained by deprotection of **199** and the benzyl group introduction in the presence of basic conditions and BnBr followed by Dess–Martin oxidation. For diastereo and regio divergent synthesis of target compounds, isomer **201** was converted to the 7*R*,8*R* isomer of princepin **203** and isomer **200** was converted to the 7*S*,8*S* isomer of princepin **202** over a few steps. (Figure 28).

### 2.3. Synthesis of Polyketides-Based Natural Products

Resorcyclic acid lactones are a major class of naturally occurring compounds [64]. Paecilomycin A-F are polyketides that belong to resorcyclic acid lactones and are isolated from mycelial culture of *Paecilomycessp*. They exhibit a number of medicinal and biological properties, such as anti-malarial, antibacterial, anti-enzymatic, anti-parasitic, nematicidal, and anti-fungal. Paecilomycins are found to express anti-plasmodial activity in opposite to *Plasmodium falciparum*. Among all paecilomycins, paecilomycin E exhibited the best activity with an IC_50_ value of 20.0 nM [65]. In 2019, Reddy et al. proposed the total synthesis of paecilomycin A-F by employing the key steps of Alder–Rickert, Mitsunobu esterification, and ring closing metathesis [66]. The total synthesis was commenced from D-ribose **204**, which was converted to alcohol **205** in a few steps. Then, the Mitsunobu reaction was performed to couple alcohol **205** and acid **206** in the presence of triphenyl phosphine (PPh_3_), diethyl azodicarboxylate (DEAD), and toluene at room temperature for 6 h to obtain esterification product **207** in an 89% yield. Further, Hoveyda Grubbs second generation catalyst was used to perform ring closing metathesis that furnished cyclized trans-macrolactone product **208** in an 80% yield. In the last step, paecilomycin E **209** was obtained in an 82% yield by employing the global deprotection of macrolactone in the presence of CH_2_Cl_2_ and BCl_3_ (Figure 29).

Styryllactones are naturally occurring organic compounds which are isolated from a number of natural sources, such as *Goniothalamus dolichocarpus* and *Cryptocarya caloneura* [67]. Since these compounds are biologically active, they have been used in Thai traditional medicine and exhibit anti-cancerous activity [68]. Kotammagari et al. proposed the total synthesis of goniothalamin and its epimers by using Ferrier reaction, Jones oxidation, acid-mediated transition metal free epimerization, and Mitsunobu reaction as key steps, in 2019 [69]. The total synthesis was commenced from triacetyl-*O*-D-glucal **210**, which was converted to compound **211** over a few steps. Next, 85% of alcohol **212** was obtained, which underwent a Mitsunobu inversion to invert stereochemistry of the OH group in the presence of *p*-nitrobenzoic acid (PNBA) and diethyl azodicarboxylate (DIAD) at 0 °C to room temperature for 12 h, followed by deprotection of the ester to obtain 60% of epimerized alcohol **213**. Further, the epimerized alcohol **213** was protected in the presence of *tert*-butyl dimethyl-silyl chloride (TBSCl) and imidazole followed by Jones oxidation and deprotection of the TBS group in the presence of BF_3_.OEt_2_ to furnish the final compound (−)-5-hydroxygoniothalamin **214** in a 30% yield along with its C-5 epimer. The compound **211** was converted to 62% of (+)-5-hydroxygoniothalamin and its C-5 epimer by oxidation in the presence of Jones reagent and deprotection in the presence of BF_3_.OEt_2_. The synthesis of (−)-5-acetylgoniothalamin **216** was accomplished by the acetylation of (−)-5-hydroxygoniothalamin **214** in an 85% yield (Figure 30). 

Limaol is a cytotoxic agent extracted from a marine dinoflagellate, *P*. *lima* [70]. Hess et al. reported the total synthesis of limaol in 2021 [71]. Stille reaction, asymmetric propargylation, a gold catalyzed-spirocyclization, Mitsunobu reaction, and substrate-controlled allylation were employed as key steps for the synthesis of limaol **223**. The synthesis commenced with the preparation of fragments **219** and **220** from commercially available starting substances, such as **217**, which were prepared from **217** and **218** over a few steps. Fragment **219** and **220** were made to couple together in the presence of MgBr_2_.OEt_2_ and dichloromethane to yield 88% of the coupled product **221**. In the next step, the Mitsunobu reaction was performed in the presence of 4-nitrobenzoic acid, triphenyl phosphine, diethyl azodicarboxylate, and toluene at 0 °C to room temperature to invert the configuration of the OH group at C-27 in compound **222** with a yield of 91%. Finally, compound **222** was converted to limaol **223** with a yield of 32% over a few steps (Figure 31). 

Petromyroxol belongs to acetogenins, which possess cytotoxic, anti-pyrexial, anti-helminthic, and anti-cancer properties, etc. [72]. Petromyroxols are isolated from a larval sea lamprey and exhibit a good olfactory response. Mullapudi et al. proposed the total synthesis of (+)-petromyroxol and its diastereomers in 2020 [73]. For the synthesis of (+)-petromyroxol, allyl glycoside **224** was made to undergo a Mitsunobu reaction in the presence of para-nitrobenzoic acid to give unsaturated compound **225** in an 81% yield. In the next step, the compound **225** was subjected to olefin oxidative cleavage followed by two-carbon homologation to give unsaturated ester derivative **226** in a 69% yield. In the later step, hydrogenation by Pearlman catalyst along with saponification of both esters yielded (+)-petromyroxol **227** in a 77% yield (Figure 32). 

For the synthesis of 6-*epi*-(−)-*iso*-petromyroxol **231**, the allyl glycoside **228** was subjected to a Mitsunobu reaction in the presence of para-nitrobenzoic acid (PNB), DIAD, and PPh_3_ at 0 °C to room temperature for 3 h to obtain the PNB-substituted product **229** in an 86% yield. In the next step, olefin oxidation of the substituted product **228** followed by two-carbon homologation gave 71% of ester **230**. The last step included Pearlman catalyst-mediated hydrogenation, followed by saponification of ester **230** to give the final product 6-*epi*-(−)-*iso*-petromyroxol **231** in a 74% yield (Figure 32). Some other diastereomers of petromyroxol were also synthesized by taking an isomer of allyl glycoside **224** as starting compounds by employing a similar strategy (Figure 33).

### 2.4. Natural Products-Based Terpenes Synthesis

Curcusones belonging to the group of diterpenes are complex natural products. They are extracted from *Jatropha curcas* and possess anti-cancerous activities. Cui et al. reported the first total synthesis of curcusone diterpenes in 2021 [74]. First of all, cyclohexenal **232** was converted to silyl enol ether, followed by a Mukaiyama aldol reaction and NaBH_4_ reduction to give **233** in a 73% yield. Further, an organocatalytic Mitsunobu reaction was performed in the presence of diethyl azodicarboxylate and alcohol **234** at 0 °C to 50 °C to yield compound **235**. The compound **235** was converted to the enone **236** over a few steps. Next, the enone **236** was methylated in the presence of KHMDS and methyl iodide to obtain two isomers (−)-*S*-curcusone A and (−)-*R*-curcusone B **238**, which were converted to (−)-*R*-curcusone C and (−)-*S*-curcusone D **240** by α-hydroxylation in 1:1 diastereomeric ratio. The compound 238 was converted to (+)-spirocurcusone **238** in a 60% yield and (−)-pyrocurcusone **239** in a 21% yield. The compound **240** was also converted to (−)-dimericursone **241** over a few steps (Figure 34). The synthesized compounds were further evaluated for anti-proliferative activity against MCF-7 cells by using a WST-1 assay. Curcusone D **240** exhibited the best cytotoxic activity among all. 

Vibsanins are diterpenoids present in the leaves of *Viburnum awabuki*, which possess piscicidal activity. They consist of 7-membered, 11-membered, and rearranged vibsanes. The total synthesis of 11-membered vibsane is reported by Takao et al. in 2020 [75]. The synthesis commenced from the preparation of fragments **244** and **245** from commercially available substances **242** and **243** over a few steps. These fragments were coupled in the presence of tertiary butyl and THF to give the compound **246** in a 52% yield. The Mitsunobu reaction between **246** and acid **247** at −15 °C was performed to afford acylate and give ester **248** in a 72% yield. Further, deprotection and oxidation of ester **248** followed by a Nozaki–Hiyama–Takai–Kishi (NHTK) reaction gave diastereoselectively pure alcohol **249** in an 82% yield. The last step involved another Mitsunobu reaction with para-nitrobenzoic acid, diethyl azodicarboxylate, and triphenyl phosphine at 0 °C, followed by chemoselective methanolysis to give the final product **250** in a 91% yield (Figure 35). 

Pavidolide B belongs to the cembranoid family, which is a pervasive class of marine natural products. These are isolated from genera *Sarcophyton* and *Sinularia*. These exhibit biological activity against human probe myelocytic cell lines. In 2019, Zhang et al. proposed the total synthesis of (−)- pavidolide B by utilizing annulation reactions and ring closing metathesis reactions as key steps to achieve a 16% overall yield [76]. The total synthesis was commenced by the reaction of unsaturated aldehyde **251** and dimethyl-2-bromomalonate **252** in the presence of 2-(diphenyl((trimethylsilyl) ocy) methyl) pyrrolidine catalyst **253** with the subsequent protection of the aldehyde group in the presence of PTSA and CH(OEt)_3_, which were selected after the screening of various conditions, followed by hydrolysis to give 80% of substrate **254** in a diastereomeric ratio of 1.5:1. In the later step, the Mitsunobu coupling with alcohol **255** was employed in the presence of diethoxyethyl azodicarboxylate (DEAD) to obtain 74% of the coupled product **256** in a similar diastereomeric ratio. After the screening of various conditions, the annulation reaction was performed in the presence of PhSH and AIBN to obtain the annulated product **257** in a 38% yield with a *dr* of 1.6:1. The compound **257** was converted to tetracyclic pavidolide **258** in a 95% yield over a few steps (Figure 36).

(+)-*ar*-Macrocarpene is a naturally occurring sesquiterpenoid derived from *Cupressus macrocarpa*. This family covers a number of biologically active derivatives. In 2019, Khatua et al. reported the total synthesis of *ar*-Macrocarpene in a 42.1% overall yield from the commercially available compound **259**, which was converted to alcohol **260** over a few steps [77]. It underwent the Mitsunobu reaction with *o*-nitrophenylsulfonyl hydrazide **261** at 0 °C for 2 h to furnish sulfonyl hydrazine **262**, which was rearranged in the presence of methanol, and gave 65% of the diazene intermediate **263** in the presence of methanol. In the final step, the diazene intermediate was further hydrogenated to yield (+)-*ar*-Macrocarpene **264** in a 99% yield (Figure 37). 

Andrographolide is known as the king of bitters as it has a lot of applications in ayurvedic and local medicine [78]. Yang et al. proposed the total synthesis of andrographolide in 2020 [79]. Diene **265** was cyclized via Diels–Alder cycloaddition by using DMAD along with hydrolysis, followed by manganese-catalyzed HAT reduction, which gave *trans*-decalin **266** in a 63% yield. It was further treated with TIPSOTf and diisobutyl aluminum hydride (DIBAL-H) to produce ene-diol **267**. The next step proceeded via a Mitsunobu reaction with 120 mol% of hydrazine derivatives in the presence of 120 mol% of diethyl azodicarboxylate and 130 mol% of triphenyl phosphine at −30 to 25 °C followed by the conversion to iodide **268** in the presence of imidazole and iodine in a 72% yield. In the next step, the treatment of iodide **268** with 2-thienyl copper lithium and vinyl bromide **269**, along with carbonylative lactonization with Palladium-Xanthophos, gave andrographolide **270** in a 92% yield (Figure 38). 

For the synthesis of 14-hydroxy-collandonin, diene **265** was converted to diol **271** over a few steps. The diol **271** underwent acylation that gave allylic carbonate **272** in a 74% yield [79]. Further, allylic carbonate **272** was reacted with 250 mol% of umbelliferone **273** via a Mitsunobu reaction in the presence of 150 mol% of diethyl azodicarboxylate and 150 mol% of triphenyl phosphine at 25 °C followed by a Tsugi reduction in the presence of palladium acetate and formic acid along with the deprotection of silyl ethers to give 14-hydroxy-colladonin **274** in a 95% yield (Figure 39).

### 2.5. Synthesis of Lactone-Based Natural Products

Lactones are naturally occurring compounds which possess various biological activities. Goniopypyrone and gonitriol are styryl lactones isolated from *Goniothalamus giganteus*. These lactones show cytotoxic activity against leukemic cells [80]. In 2019, Miyazawa et al. proposed the total synthesis of (+)-goniopypyrone and (+)-goniotriol by using key steps involving Pd-catalyzed carbonylation and diastereoselective reduction of ynone [81]. The total synthesis was commenced from D (−)-tartaric acid **275**, which was converted to ynone **276** in a few steps. A number of reagents were screened out for diastereoselective reduction of ketonic group in ynone **276** and (*S*)-CBS catalyst and borane dimethyl sulfide catalyst were used to yield 12:88 ratio of diastereoselective alcohol **277** in a 12% yield and **278** in an 88% yield. Undesired diastereoisomer **277** was converted to **278** via the Mitsunobu reaction in the presence of para-nitrobenzoic acid (*p*-NBA) and diethyl azodicarboxylate (DEAD) in a 74% yield over two steps. The next step included the protection of the hydroxyl group of **278**, followed by iodide introduction in the presence of NIS along with NsNHNH_2_-mediated diimide reaction to give 96% of **279**. The compound **279** underwent deprotection of acetonide followed by Pd-catalyzed carbonylation in the presence of 5 mol% of Cl_2_Pd (PPh_3_) and Et_3_N, with subsequent deprotection of benzyl ether along with DBU treatment to furnish 73% of (+)-goniopypyrone **280** (Figure 40).

Furthermore, the total synthesis of (+)-goniotriol was commenced from alcohol **281**, which was subjected to the Mitsunobu reaction in the presence of para-nitrobenzoic acid (*p*-NBA) and diethyl azodicarboxylate (DEAD) to yield 91% of **282 [81]**. In the next step, the iodide group was introduced in the presence of NIS in AgNO_3_ with a subsequent diimide reaction in the presence of NsNHNH_2_, followed by the deprotection of the acetonide group in acidic conditions to yield 91% of the cyclized precursor **283**. In the last step, Pd-catalyzed carbonylation gave a 93% yield, followed by the deprotection of benzyl ether in the presence of TiCl_4_ to give 66% of (+)-goniotriol **284** (Figure 41).

Epi-zeaenol and zeaenol are lactones isolated from filamentous fungi. They are a biologically active natural product. Zeaenol expressed anti-cancerous activity against human cancer cell lines, with an IC_50_ value of >50 µM and inhibition against NF-ĸB cells [82]. The total synthesis of epi-zeaenol and zeaenol was reported by Doda et al. in 2019 [83]. Mitsunobu inversion, Heck cross coupling reaction, De Brabander’s protocol for macrolactonisation, Ohira–Bestmann alkynylation, and alkyne aldehyde coupling are the key steps involved in the synthesis of zeaenol to obtain a 21% overall yield. The total synthesis of zeaenol was commenced from D-mannitol **285**, which gave macrolactone **286** over a few steps. On treatment with TMSCl in methanol, macrolactone **286** was converted into allyl alcohol **287** in an 86% yield. The next step involved the Mitsunobu reaction in the presence of 4-nitrobenzoic acid, DEAD, and PPh_3_ at room temperature for 12 h to invert the OH group followed by ester methanolysis to obtain alcohol **288** in an 81% yield over two steps. The alcohol **288** was treated with an excess of TiCl_4_ to obtain zeaenol **289** in an 83% yield (Figure 42).

Resorcyclic acid lactones, also called secondary polyketides, are extracted from fungal strains, i.e., *Lasiodiplodia Theobromae*, *Penicillium* sp., *Syncephalastrum racemosum*, etc. They exhibit a number of medicinal properties, such as anti-malarial, anti-microbial, and anti-cancerous, etc. [84]. Various research groups have reported the total synthesis of macrocyclic lactones except penicimenolide A. Das et al. reported the total synthesis of penicimenolide A, *R*-resorcyclide, and *R*-dihydroresorcyclide in 2021 by employing ring closing metathesis as key step [85]. The total synthesis was started from commercially available epoxide fragment **290**, which gave the acid fragment **291** over a few steps that further coupled with hexenol fragment **292** via the Mitsunobu reaction in the presence of PPh_3_, DIAD, and THF at 0 °C to room temperature to give compound **293**. The oxidation of compound **293** by Grubb’s II oxidation method and deprotection with K_2_CO_3_ in methanol provided the cyclized product **294**. Further, the OH group in **294** was oxidized by Dess–Martin periodinane and demethylated by using AlI_3_ and phloroglucinol to give (*R*)-penicinomenolide **295** in a 34% yield. The cyclized compound **294** was converted to (*R*)-dihydroresorcyclide **296** in a 61% yield over a few steps (Figure 43).

The synthesis of trans-resorcyclide **302** was commenced from a benzaldehyde derivative **297**, which was converted to an acid **298** [85]. The next step involved the Mitsunobu reaction to couple compound **298** with 2-hydroxy heptene **299** to give a 62% yield of macrocycle **300**. Further, Grubb’s II oxidation was performed followed by deprotection and Dess–Martin periodinane-mediated oxidation to give cyclized product **301** in a 60% yield. In the final series of steps, the OH group was first masked by selenium protection and then demethylated using AlI_3_ and phloroglucinol. Selenide was demethylated and oxidized to obtain the final product (*R*)-trans- resorcyclide **302** in a 54% yield (Figure 44). 

Pyrenophorins are isolated from *Pyrenophora avenae* and exhibit anti-bacterial and anti-fungal properties, etc. Edukondalu et al. proposed the total synthesis of (−)-pyrenophorin in 2020 [86]. The synthesis commenced from epoxy bromide **303**, which was alkylated with 2-vinyl-1,3-dithiane **304**, followed by LAH-mediated epoxide opening and imidazole-mediated TBSCl protection of the obtained alcohol to provide 82% of olefin **305**. The ozonolysis of olefin **305**, accompanied by treatment with (methoxycarbonyl methylene) triphenyl phosphorane, and hydrolysis, along with desilylation, gave hydroxy acid **306** in an 86% yield. In the next step, dimerization of hydroxy acid **306** via the Mitsunobu reaction by employing diethyl azodicarboxylate, triphenyl phosphine, and a 10:1 ratio of toluene and THF provided a 61% of dimer **307**. The deprotection of 1,3-dithiane group with CaCO_3_ gave the desired product pyrenophorin **308** in a 73% yield (Figure 45). 

#### 2.5.1. Synthesis of Glutamate Receptors

Glutamate is an excitory neurotransmitter in mammalian cells. It acts on a number of receptors and regulates important brain functions. The dysfunction of these receptors leads to many abnormalities. These receptors are activated by a number of positive allosteric modulators (PAMs). Yamada et al. reported the total synthesis of some positive allosteric modulators (PAMs) **313**, **316**, **319**, and **322** for mGlu_2_ and mGlu_3_ receptors, in 2021 [87]. Starting with butanone derivative **309**, alkylation was performed by employing either –*R* or *S*-oxirane derivative **310** to obtain **311** in a 97% yield. In the next step, the Mitsunobu reaction with methyl-4-methoxy-5-hydroxy benzoate **312** and di-*tert*-butyl azodicarboxylate at 120 °C for 45 min gave a 65% yield, which was further saponified under microwave irradiation at 100 °C for 30 min to give the final product **313** in an 87% yield (Figure 46). 

For synthesis of PAMs **316**, 2-flouro-benzoate **314** was used as a starting material and reacted with methyl oxiranes under a microwave temperature of 350 °C for 3 h to provide compound **315** in a 97% yield [86]. Then, the Mitsunobu reaction with methyl-4-methoxy-5-hydroxy benzoates **312** at 120 °C for 45 min, along with saponification under microwave irradiation of 100 °C for 30 min, yielded 87% of the compound **316** (Figure 47).

To obtain compound PAMs **319**, butanone derivative **309** was alkylated with -*R* or -*S*-3-bromo-2-methoxy propan-1-ol **317** in the presence of K_2_CO_3_, KI, and ACN at 60 °C to give 69% of compound **318**. Further, it underwent for the Mitsunobu reaction with methyl-3-methoxy-4-hydroxy benzoate **312** at 120 °C for 45 min, and was then saponified by microwave at 100 °C for 30 min, to obtain **319** in a 30–43% yield (Figure 48).

The synthesis of PAMs **322** was commenced from **309**, which was alkylated with bromo propanol **317** at 60 °C to obtain compound **320** in an 87% yield. A total of 20–55% of the final product **322** was obtained by employing the Mitsunobu reaction with phenol in the presence of di-tert-butyl azodicarboxylate and triphenyl phosphine at 120 °C for 45 min, followed by saponification at 100 °C for 30 min (Figure 49). 

The synthesized positive allosteric modulators were selected for the evaluation of their biological activities. These receptors were tested for their potential to act on human mGlu_2_ and mGlu_3_ cell lines. In the case of PAMs **322**, the best activity is observed by substituting 4-hydroxy-2-methoxy benzoic acid that exhibited 0.028 µM against mGlu_2_ cells, and 0.512 µM against mGlu_3_ cells. 

#### 2.5.2. Synthesis of Steroid-Based Natural Products

Steroids are an important class of natural compounds possessing a number of biological activities. Dankasterones A and B and periconiastone A are steroids that are isolated from *Gymnacella dankaliensis* and *Periconia* sp., respectively, and exhibit anti-cancerous and anti-bacterial properties. Dankasterone A demonstrate anti-cancer potential against MG-MID-5.41. Dankasterone A and B exhibited marginal and significant growth inhibition against murine P388 cells. Furthermore, periconiastone A manifested significant anti-bacterial activity against Gram-positive *E*. *faecalis* and *S*. *aureus* with MIC values of 32 and 4 µg/mL [88]. In 2020, Chen et al. reported the total synthesis of dankasterones A and B and periconiastone A by employing metal-catalyzed hydrogen atom transfer and position selective C-H oxygenation as key steps [89]. The total synthesis was accomplished in 22–23 steps. The synthesis started with ketone **323**, which, over a few steps, gave fragment **324**. The compound **325** was converted to fragment **326** over a few steps, which, coupled with **324**, in the presence of *tert*-BuLi followed by oxidation gave the coupled product **327** in 67% over two steps. The coupled product **327** was converted to compound **328** over a few steps, which underwent deprotection of ketal in the presence of *p*-TsOH, LAH-mediated reduction and a Mitsunobu reaction in the presence of di-*tert*-butyl-azodicarboxylate and triphenyl phosphine to yield olefin **329** in 82% over three steps. The oxidation of olefin **329** took place in the presence of CrO_3_ using 3,5-DMP as promotor and HClO_4_-supported C-H oxygenation to obtain spirocycle **330** in a 67% yield. Further, the rearrangement of **330** by deacetylation followed by oxidation with DMP and Julia–Kocienski olefination with **331** gave dankasterone B **332** in a 35% yield. Dehydrogenation of **332** in the presence of PhI(OH)(OTs) gave dankasterone A **333** in a 28% yield. The intramolecular aldol reaction of **332** by treatment with DBU in the presence of TBSOTf gave periconiastone A **334** in a 46% yield (Figure 50). 

#### 2.5.3. Synthesis of Chromone-Based Natural Products

Preussochromone belongs to a family of chromone natural products which have a tricyclic skeleton and a chromenone core with a dihydrothiopyran ring. It shows biological activity against human cancer cell lines with an IC_50_ value of 25.5 µM against HeLa and an IC_50_ value of 8.34 µM against A549 cells [90]. Beller et al. reported the total synthesis of preussochromone A in 2020 [91]. The significant steps employed for the synthesis of (−)-preussochromone are Lewis-acid-mediated cycloisomerization and thia-Michael-retro-Michael addition reaction that provided the target compound in an 8% overall yield in 11 steps. The synthesis started from a primary alcohol **335**, which was converted to a 71% yield of thiobenzoate **336** by employing the Mitsunobu reaction in the presence of 2 eq. of BzSH, 1.5 eq. of diazo dicarboxylate, and 1.5 eq. of triphenyl phosphine at 0 °C to room temperature. Dihydroxylation of thiobenzoate **336** along with thioester cleavage in the presence of sodium methoxide gave diol **337**. It was further coupled with 2-sulfonyl chromenone **338** via thia-Michael-*retro*-Michael addition in the presence of K_2_CO_3_ to furnish diol **339** in a 94% yield. The diol **339** was converted to the target compound preussochromone A **340** over a few steps providing a final yield of 61% (Figure 51). 

#### 2.5.4. Synthesis of Acetylenic-Based Natural Products

Asparenydiol consists of acetylene in its structure and is isolated from *Asparagus Officinalis*. It possesses many biological properties, such as anti-cancer, anti-pyretic, anti-diabetic, and immunomodulators. It exhibits anti-proliferative activity against Col-2, KB, LNCap, HU-VEC, and Lu-1 cell lines in the range of 4–20 µg/mL [92]. Casotti et al. reported the total synthesis of asparenydiol in 2020 by employing the Mitsunobu reaction and Sonogashira cross-coupling reaction as key steps to furnish it with a 30% overall yield [93]. The synthesis commenced with iodophenol **341**, which was converted to terminal alkyne **342** over a few steps. It was coupled with methyl (*E*)-3- methyl bromo acrylate **343** via Sonogashira coupling by taking Pd-Cu/PVPy as catalyst and 2,2,6,6-tetramethylpiperidine as base to give a 57% yield of **344**. The terminal acetate group underwent DIBAL-mediated reduction at a low temperature to give 90% of primary alcohol **345**. A Mitsunobu reaction for the etherification of **345** with phenol derivative **346** in the presence of diethyl azodicarboxylate and triphenyl phosphine at 0 °C to 20 °C for 6 h provided an isolated yield of 84% followed by the deprotection of the hydroxyl groups by treatment with TBAF to give the target compound asparenydiol **347** in a 30% yield (Figure 52).

#### 2.5.5. Synthesis of Piperidinones-Based Natural Products

Pipermethystine is a secondary alkaloid present in *Piper Methysticum*, which possess many biological properties [94]. Vázquez-Amaya et al. reported the total synthesis of pipermethystine by employing the Mitsunobu reaction in 2020 [95]. Starting from hydroxy (*R*)-hydroxypiperidine **348**, triple C-H functionalization gave piperidone derivative (*R*)-**349** in a 54% yield. In the next step, the Mitsunobu reaction with AcOH in the presence of diethyl azodicarboxylate (DIAD) and triphenyl phosphine (PPh_3_) followed by deprotection with ceric ammonium nitrate (CAN) gave 70% of ester (*S*)-**350**. The acylation of ester (*S*)-**350** with cinnamoyl chloride provided (*S*)-pipermethystine **351** in a 72% yield. The configuration of the synthesized compound was opposite to the naturally occurring isomer. For that reason, piperidone derivative **349** was *O*-acetylated in a basic condition followed by oxidative debenzylation and treatment with cinnamoyl chloride for *N*-acylation to get (*R*)-pipermethystine **352** in a 70% yield (Figure 53).

#### 2.5.6. Synthesis of Peptide-Based Natural Products

Asperipin is a biologically important compound. It is extracted from *Aspergillus flavus*. Shabani et al. reported the total synthesis of aspirin in 2020 [96]. In the first step, (*R*)-glyceraldehyde acetonide **353** was treated with phenyl magnesium bromide to give alcohol **354** in an 85% yield with 99:1 diastereomeric ratio. It was further treated with tyrosine derivative **355** via Mitsunobu reaction in the presence of DIAD and PPh_3_ under sonication to obtain ether adduct **356** in a 45% yield. The nosylation of adduct **356** in the presence of KHMDS at −78 °C, along with the hydrolysis of the ester and oxazolidine groups, in the presence of Cs_2_CO_3_ and LiOH yield 85% of the β-hydroxy tyrosine derivative **357**. The β-hydroxy derivative **357** was coupled with Thr(Bn)-Gly-O*t*Bu followed by a Mitsunobu reaction in the presence of DIAD and PPh_3_ to give aziridine **358** in a 90% yield. Over a few steps, the aziridine **358** was converted to the desired product asperipin-2a **359** (Figure 54). 

#### 2.5.7. Synthesis of Amino Acids-Based Natural Products

The natural compounds containing 1,3-diols are among the most important organic compounds, which possess many biological and medicinal properties. Galantinic acids are natural amino acids containing 1,3-diols and act as the back-bone of galantin, an antibiotic. 1-Deoxy-5-hydroxy-sphingolipid also belongs to this class of compounds and acts as anti-prostate cancer substances [97]. Rehman et al. reported the total synthesis of (−)-galantinic acid and hydroxy-sphingolipids in 2020 [98]. The total synthesis of (−)-galantinic acid commenced from (±)-glysidyl benzyl ether **360**, which was converted to intermediate compound **361** over a few steps. In the next step, regioselective epoxide ring opening was performed in the presence of benzyl alcohol and catalytic amount of BF_3_.OEt_2_ to obtain compound **362** in a 93% yield. Further, the Mitsunobu reaction was employed for the conversion of the secondary hydroxyl group to azide **363** with mesylate chloride and sodium azide at 80 °C for 7 h in an 88% yield. The last step included the introduction of carboxylic group in the presence of sodium periodate with RuCl_3_.6H_2_O as a catalyst followed by Pd-catalyzed hydrogenation in MeOH to furnish (−)-galantinic acid **364** in an 89% yield (Figure 55).

The synthesis of 1-deoxy-5-hydroxysphingolipid **367** started from an epoxide intermediate **361**, which was reduced in the presence of DIBAL-H, followed by oxidation in the presence of 1-dodecene with 5 mol% of Grubb’s catalyst to give **365** in a 72% yield [98]. The next step was performed via the Mitsunobu reaction in the presence of mesylate chloride and triethyl amine with sodium azide at 80 °C for the conversion of the secondary hydroxyl group to azide **366** in an 85% yield. Lastly, azide **366** was reduced in the presence of 10% Pd on C and hydrogen to yield target compound 1-deoxy-5-hydroxysphingolipid **367** in a 91% yield (Figure 56).

Jomthonic acid A is an amino acid derivative isolated from the genus *Streptomyces* of the soil-derived actinomycete. It expresses anti-atherogenic and anti-diabetic activity against S-13 preadipocytes of mice. It also acts as an inhibitor of preadipocytes differentiation into mature adipocytes in the 2–50 µM range [99]. It has many structural features that were built by employing the Mitsunobu reaction, Gilman reaction, Yamaguchi esterification, and amide coupling. Dumpala et al. reported the total synthesis of jomthonic acid A in 2019 [100]. The synthesis was initiated from *trans*-cinnamyl alcohol **368**, which was epoxidized by employing sharpless asymmetric epoxidation in the presence of (+)-DIPT and Ti(O*^i^*Pr)_4_ followed by regioselective ring cleavage with Gilman reagent to give diol **369** in a 68% yield. The diol **369** was subjected to selective hydroxyl protection in the presence of TBSCl/Bu_2_SnO/CH_2_Cl_2_ in imidazole providing a 93% yield of silyl ether **370**, leading towards the Mitsunobu reaction with diphenyl phosphorazidate (DPPA) and DIAD at 0 °C to rt to obtain azide **371** in an 85% yield. Deprotection of azide **371** was performed in the presence of TBAF, followed by oxidation in the presence of TEMPO and PhI(OAc)_2_ in CH_3_CN/H_2_O, to give acid fragment **372** in a 93% yield. The fragment (R)-**374** was synthesized from (3*R*)-3-hydroxybutanoate **373** over a few steps. In the next step, acid fragment **372** and silyl ether **374** were coupled via Yamaguchi esterification in the presence of 2,4,6-trichlorobenzoyl chloride to give 68% of azide **375**. The jomthonic acid **376** was obtained from azide **375** in an 80% yield over a few steps (Figure 57). 

#### 2.5.8. Synthesis of Natural Products Containing Carbohydrate Derivatives

Ampelomycins are polyhydroxylated cycloalkane-type compounds, also known as “carbohydrate mimetics”, and are isolated from a fungal strain *Ampelomyces* sp. [101]. They can act as important scaffolds for the synthesis of drugs and analogues. In 2019, Brindisi et al. reported the total synthesis of ampelomins [102]. The main steps employed for the synthesis of ampelomins were the Mitsunobu reaction, stereoselective hydrogenation, and a regioselective and stereoselective nucleophilic ring opening. The synthesis of ampelomin B was initiated from diol **377**, which underwent diol protection with DMP, followed by halohydrin formation in the presence of iodine and AcOAg, along with deprotection of diol at 40 °C and ring closure with DBU at room temperature to give β-epoxide **378** in a 99% yield. The homogenous hydrogenation of epoxide **378** with 10 mol% of Crabtree’s catalyst at room temperature gave 99% of alcohol **379**. Further, the Mitsunobu reaction was performed in the presence of para-nitrobenzoic acid, diethyl azodicarboxylate, and benzene for 8 h to obtain 47% of the substituted product **380**. Methanolysis of **380** gave the desired (−)-ampelomin B **381** in an 83% yield (Figure 58).

The synthesis of ampelomin D started from diol **377**, which was converted to substituted diol **382** over a few steps [102]. The next step involved the inversion of the configuration via the Mitsunobu reaction in the presence of *p*-nitrobenzoic acid, diethyl azodicarboxylate, triphenyl phosphine, and benzene under reflux to afford **383** in a 39% yield. Further, the hydrolysis of **383** at room temperature gave the final product (+)-ampelomin D **384** in a 96% yield with an overall yield of 14% (Figure 59). 

#### 2.5.9. Synthesis of Natural Products-Based Fatty Acid Amides

Serinolamine A and columbamide D are endocannabinoids, which are extracted from a cyanobacteria, *Lyngbya majuscula.* These exhibit many biological properties as they act as anti-proliferatives against lung cancer and breast cancer cells [103,104]. Ghotekar et al. reported a concise and efficient total synthesis of serinolamides and columbamides in 2019 [105]. The total synthesis commenced from the preparation of two fragments, **386** and **388**. The fragment **386** was synthesized from tetradecanol **385** over a few steps. 9-Decen-1-ol **387** was converted to fragment **388** over a few steps. Next, (*R*)-benzyl glycidyl ether **389** underwent regioselective opening under basic conditions to give (*R*)-triol **390** in a 98% yield. In the next step, the Mitsunobu reaction was performed for amine linkage formation in the presence of diethyl azodicarboxylate, triphenyl phosphine, and DMF at 0 °C for 6 h to obtain (*S*)-amino alcohol **392** in a 95% yield. Further, nosyl deprotection in the presence of thiophenol in acetonitrile, followed by hydrogenation with H_2_/Pd(OH)_2_, gave enantiopure alcohol (*R*)-**393** in a 97% yield. The alcohol (*R*)-**393** was used as a precursor to couple the fragment **388** for the synthesis of columbamide D **394** in a 71% yield. Similarly, alcohol (*R*)-**393** was coupled with fragment **386** to furnish serinolamide-A **395** in a 75% yield (Figure 60).

#### 2.5.10. Synthesis of Chlorosulpholipids-Based Natural Products

Chlorosulfolipids are complex marine natural products extracted from *Ochromonas danica*, a freshwater algae [106]. These natural products possess antimicrobial and antiviral properties and are integral parts of membranes. Sondermann et al. proposed the total synthesis of mytilipin B in 2019 [107]. The total synthesis was achieved after solving many problems regarding deviations of the spectrum of naturally occurring and laboratory synthesized mytilipin. The total synthesis was commenced from epoxide **396**, which was converted to 1,2-diol **397** over a few steps. The 1,2-diol **397** underwent 1,4-dioxolane formation in the presence of copper sulfate and *p*-toluene sulfonic acid, followed by hydrogenolysis with Pd/C and ethyl acetate, leading to the Mitsunobu reaction with 1-phenyl-1*H*-tetrazole-5-thiol, triphenyl phosphine, and THF at 0 °C to 23 °C to get a quantitative yield of **398**. The compound **398** was converted to sulfone **399** in an 84% yield by oxidation in the presence of (NH_4_)_6_Mo_7_O_24_.4H_2_O and hydrogen peroxide. The sulfone **399** was converted to 4a diastereomer **400** of mytilipin B in a 72% yield. The sulfone **399** was also converted to 4b diastereomer of mytilipin B **401** in a 72% yield by adopting a different route over a few steps (Figure 61).

#### 2.5.11. Synthesis of Dihydropyran-Based Natural Products

Marine organisms are a source of many natural products. Eurotiumide F and G are marine fungi-derived natural products, which possess many biological properties, i.e., antimicrobial and antifouling. Nakayama et al. reported the total synthesis of eurotiumide F and G in 2019 [108]. Aldehyde **402** and 1-phenyl-1*H*-tetrazole-5-yl (PT)-sulfone **403** were reacted together to give ketone **404** over a few steps. It underwent hydrolysis in the presence of KOH in ethanol under reflux, followed by an intramolecular Mitsunobu reaction in the presence of diethyl azodicarboxylate and triphenyl phosphine in THF to obtain *trans*-4-methoxyisochroman-1-one **405** in an 80% yield over two steps. The last step involved the reduction with DIBAL to obtain lactol with the subsequent removal of the MOM group to obtain the final compound (−)-eurotiumide G **406** in a 40% yield. The compound *trans*-4-methoxyiso-chroman-1-one **405** was converted to (+)-eurotiumide F **407** in a 46% yield over a few steps. (+)-Eurotiumide G **408** was synthesized from ketone **404** by DIBAL-mediated reduction and MOM deprotection by treatment with 6 M HCl in methanol in a 55% yield (Figure 62). 

#### 2.5.12. Synthesis of Enediynes-Based Natural Products

Kedarcidin chromophore is an important natural product with biological properties, such as antitumor and antibacterial, etc. It exhibits selective in vivo antitumor activity against B16 melanoma cells and P388 leukemia [109]. It was discovered for the first time in 1990. The first total synthesis of kedarcidin chromophore, comprised of 17 steps, was proposed by Lear et al. in 2019 by employing a stereoselective allenyl zinc keto addition, Mitsunobu etherification, α-selective glycosylations, atropselective Sonogashira and Shiina cyclization sequence, Ce-mediated enediyne cyclization, and Ohfune-based amidation [110]. The total synthesis was commenced from ketone **409**, which was converted to compound **410** over a few steps. A total of 1.1 Eq. of β-amino-2-chloroazatyrosine **411** was utilized for the inversion of allylic C 11-β alcohol via Mitsunobu reaction in the presence of di-2-methoxyethyl azodicarboxylate, triphenyl phosphine, and toluene at 0 °C to deliver compound **412** in a 73% yield. The compound **412** was converted to kedarcidin chromophore **413** over a few steps. (Figure 63). 

### 2.6. Synthesis of Epidithiodiketopiprazine-Based Natural Products

The importance of the Mitsunobu reaction is evident from its role in the total synthesis of natural products. Scabrosins are ETPs (epidithiodiketopiperazine) which are fungal metabolites with a diverse structure and biological significance as antineoplastic and cytotoxic substances. They express in vitro cytotoxic activity against human breast cancer cells (MCF-7) with an IC_50_ value of 1 nM, and murine leukemia P388 with an IC_50_ value of 16 nM. Scabrosins also exhibit appreciable in vivo antineoplastic activity [111]. These factors make scabrosins an important scaffold for carrying out total synthesis. Due to their complex bonding, the total synthesis of these natural products has not been reported yet. In 2019, Liu et al. reported the synthesis of the desulfur-scabrosin skeleton, common to most ETPs, by using the Mitsunobu reaction, pyrrolidine ring construction, asymmetric nucleophilic epoxidation, and base-induced keto-enol isomerization as key steps [112]. The synthesis was started from phenol **414**, which was converted to epoxycyclohexene alcohol **415** over a few steps. The epoxycyclohexene alcohol **415** was subjected to Mitsunobu coupling with compound **416** in the presence of diethyl azodicarboxylate and triphenyl phosphine to afford compound **417** in a 91% yield. The compound **417** was converted to ester **418** over a few steps, which underwent saponification with lithium hydroxide followed by dimerization in the presence of HATU/DIPEA and deprotection with TBAF to yield a quantitative amount of diastereomer b` of the desulfur-scabrosin skeleton **419**. The diastereomer b` was converted to diastereomer c` of the desulfur-scabrosin skeleton **420** in an 80% yield by base-mediated keto-enol isomerization in the presence of NaOMe in MeOH, followed by desilylation with TBAF (Figure 64). 

### 2.7. Liponucleoside-Based Natural Product Synthesis

Caprazamycins are liponucleoside-based natural products consisting of uridine attached to some amino acids. It has a diazepanone ring in its structure, which is a common structure in liponucleosides. These are obtained from *Streptomyces* sp. MK730-62F2. Caprazamycins exhibit biological activity as antimicrobial agents and provide a powerful ingredient for the synthesis of synthetic derivatives and drugs [113]. Nakamura et al. proposed the total synthesis of caprazol and caprazamycin A by utilizing an intramolecular Mitsunobu reaction as a key step in 2019 [114]. The total synthesis commenced from the preparation of 59% of diethyl isocyanomalonate **422** from diethylaminomalonate **421** in the presence of triphosgene and activated charcoal in 1,4-dioxane. In the next step, the aldol reaction between aldehyde **423** and diethyl isomalonate **422** was conducted in the presence of (*S*, *S*)-thiourea catalyst **424** to furnish oxazolidine **425** in a 42% yield. Further, oxazolidine **425** was converted to compound **426** over a few steps, which underwent the Mitsunobu reaction in the presence of di-*tert*-butyl azodicarboxylate, triphenyl phosphine, and toluene to afford 75% of **427**. The compound **427** was converted to protected caprazol **428** over a few steps, which underwent Troc group removal and reductive amination followed by Pd black treatment in methanol to furnish caprazol **429** in a 46% yield. The caprazamycin A **430** was obtained from protected caprazol **428** in a 90% yield over a few steps (Figure 65). 

#### Miscellaneous

Aspongodopamine A and B are heterocyclic natural products extracted from *Aspongopus chinensis*. They have many applications in the field of science and are used as pain-killers and also as food. The total synthesis of aspongodopamine has been reported by Ding et al. in 2020 [115]. A known alcohol **431** was converted to a secondary alcohol **432** over a few steps. In the next step, the secondary alcohol **432** was subjected to nucleophilic substitution with ester **433** followed by the addition of sodium azide in TBAI to provide azide **434** in a 92% yield. Further, the Mitsunobu reaction was performed with bis-boc adenine **435** in the presence of diisopropyl azodicarboxylate in THF to afford **436** in a 70% yield. Finally, 56% of aspongodopamine B **437** was furnished from compound **436** over a few steps (Figure 66).

The process of synthesis of natural products is very complicated and involves a number of strategies to achieve final compounds. In that regard, the Mitsunobu reaction plays an important part. Tetrodotoxin is a marine natural product with a wide range of origins. It is a highly complex natural product whose total synthesis is still unknown, but the synthesis of its naturally occurring intermediates Cep-212 and Cep-210 were reported by Adachi et al. in 2019 [116]. The synthesis of Cep-212 **443** started from geraniol **438**, which was converted to alcohol **439** over a few steps. In the next step, the azide group of **439** was reduced, followed by guanidinylation with *N*, *N*`-bis-boc-*S*-methylisothiourea **440** in the presence of mercuric chloride and triethyl amine to afford di-Boc guanidine **441** in a 58% yield. The next step involved the Mitsunobu reaction in the presence of DIAD and PPh_3_ to give cyclic guanidine **442** in a 59% yield. Finally, the deprotection of protecting groups was performed in the presence of K_2_CO_3_ in MeOH, followed by treatment with TFA, to yield 60% of the final product **443** Cep-212 (Figure 67). 

The synthesis of Cep-210 commenced from intermediate **439**, which was converted to allylic alcohol **444** over a few steps. In the following series of steps, allylic alcohol was protected in the presence of TBSOTf, followed by azide hydrolysis with lithium aluminium hydride and gunidinylation with *N*, *N*`-bis-boc-*S*-methylisothiourea **440** of subsequent amine, to give di-Boc guanidine **445** in an 88% yield over two steps. Further, the Mitsunobu reaction was performed under classical conditions of DIAD and PPh_3_ to obtain cyclic guanidine **446** in an 87% yield. The deprotection of the TBS and Boc group was performed in the presence of TFA in CH_2_Cl_2_, along with treatment with silica gel, to yield the final product Cep-210 **447** in a quantitative yield (Figure 68).

Cladosporin is a secondary metabolite that is isolated from various fungi, i.e., *Aspergillus flavus* and *Cladosporium cladosporioides*. It possesses many biological properties, such as antifungal, antibacterial, insecticidal, and plant growth inhibitory, etc. It was found to exhibit potent antiparasitic activity with an IC_50_ value of 40 nM against liver and blood stage proliferation of pathogens. In addition, cladosporin was found to be >100 times more selective against parasitic KRS. It is found to inhibit KRSs from *Loa Loa* and *Schistosomamansoni* species [117]. Das et al. reported the total synthesis of cladosporin in 2019 [118]. The total synthesis commenced from (*S*)-propylene oxide **448**, which was converted to intermediate **449** over a few steps. The intermediate **449** was epoxidized in the presence of m-CPBA to yield epoxide **448** in an 88% yield. Next, the Grignard reaction was performed in the presence of 1-bromo-3,5-dimethoxy benzene **449** to give two diastereomers **452** and **453** in 1:1 diastereomeric ratio with an 86% overall yield. The diastereomer **452** was converted to **453** via the Mitsunobu inversion in the presence of *p*-nitrobenzoic acid, diisopropyl azodicarboxylate, and triphenyl phosphine in THF, followed by ester hydrolysis with K_2_CO_3_ in MeOH, to give alcohol **453** in an 83% yield over two steps. The alcohol **453** was iodized with NIS in CH_2_Cl_2_, followed by treatment with Pd(PPh_3_)Cl_2_ and 1,10-phenanthroline along with demethylation, to obtain the final product cladosporin **454** in a 61% yield (Figure 69). 

The structure determination along with total synthesis of rarely occurring natural products is a complex and tedious process. Chloroenyne and related natural products are obtained from *Laurencia majuscule* [119]. Its structure has been determined by computational methods and NMR analysis of its diastereomers. Shepherd et al. proposed the total synthesis of four different diastereomers of chloroenyne and the total synthesis of notoryne in 2019 [120]. The total synthesis was commenced from epoxy alkene **455**, which was subjected to ozonolysis in the presence of ozone, PPh_3_, and NaBH_4_, followed by the Mitsunobu reaction with 1-phenyl-1*H*-tetrazole-5-thiol **456**, DIAD, and PPh_3_, followed by treatment with 3-chloroperbenzoic acid **457**, to furnish tetrazole sulfone **458** in a 60% yield. Epoxide **459** was converted to aldehyde **460** over a few steps. Next, tetrazole sulfone **458** and aldehyde **460** were coupled together in the presence of NaHMDS and DME to give **461** in a 80% yield. The **461** was converted to alkene **462** over a few steps. The alkene **462** was reacted with crotonaldehyde in the presence of Grubb’s second-generation catalyst for a cross-metathesis reaction, followed by treatment with trimethylsilyl diazomethane and the deprotection of the *p*-methoxy benzyl group, to give diastereomer c **463** of chloroenyne in an 84% yield. The diastereomer c **463** underwent the Mitsunobu inversion with *para*-nitrobenzoic acid **464**, DIAD, and PPh_3_, along with ester methanolysis to give diastereomer a **465** in a 25% yield over two steps (Figure 70). 

Furthermore, alcohol **466** underwent the Mitsunobu inversion in the presence of DIAD, PPh_3_, and para-nitrobenzoic acid **464** followed by ester methanolysis to afford inverted alcohol **467** in a 79% yield over two steps. In the next step, enyne introduction was performed by the reaction of alcohol **467** with crotonaldehyde in the presence of Grubb’s II catalyst to give diastereomer b **468** in a 39% yield over three steps. Diastereomer d **469** was prepared from **468** by the Mitsunobu inversion in the presence of DIAD, PPh_3_, and *p*-nitrobenzoic acid **464** followed by ester methanolysis (Figure 71).

Further, synthesis of notoryne **472** was commenced from chloride **462**, which was deprotected by BCl_3_.SMe_2_, followed by the Mitsunobu inversion in the presence of DIAD, PPh_3_, and *para*-nitrobenzoic acid **464** to give alcohol **470** in a 91% yield. Terminal alkene in **470** was subjected to ozonolysis followed by the Yamamoto–Petersen reaction to give (Z)-enyne **471** in a 32% yield with high diastereoselectivity. The final step involved the deprotection of (Z)-enyne **471** to give diastereoselective (Z)-notoryne **472** in a 95% yield (Figure 72). 

## 3. Conclusions

To conclude, the Mitsunobu reaction has been in use for the past few decades. Throughout this review, the significance of the Mitsunobu reaction in the total synthesis of natural products from recent years has been mentioned and explained. The smooth inversion of configuration, short reaction time, mild reaction conditions, easy separation, and purification techniques are some of the salient features achieved by employing the Mitsunobu reaction. It has adopted a wide range of pronucleophiles that expand from alcohols, carboxylic acids, imides, and sulfonamides to β-ketoesters, purines, and oximes, allowing the formation of C-C, C-N, and C-O bonds. As a lot of research has been done regarding the Mitsunobu reaction, there is still space present in this area to improve the efficacy of this reaction. With the progress of time, innovative methods should be investigated and utilized for large scale synthesis of new classes of products.

## Data Availability

Not applicable.

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
