# Peer review of "Mitsunobu Reaction: A Powerful Tool for the Synthesis of Natural Products: A Review"

_molecules, 2022, doi:10.3390/molecules27206953_

Round 1

Reviewer 1 Report

Pags. 3 – 4

Line 89 - Scheme 3:

- phenanthracene 12  is the acid/pronucleophile in the Mitsunobu reaction ? Reaction alcohol x phenol ?

Pag. 4

Line 98: .......which wasextracted from a.......  Change to  ......... which was extracted from a......

Pag. 4

Line 108: ...... presence of .3 mol% of ........ :   .3 ???

Pag. 71, Line 1224:

........diastereoselective (Z)-notoryne 470 in 95% yield        Change to     ......diastereoselective (Z)-notoryne 472 in 95% yield (relative to 471).    

Author Response

Dear Reviewer,

Thank you very much for peer reviewing our manuscript and we appreciate your complimentary recommendations as your comments have helped us significantly to improve the manuscript. We have carefully scrutinized the suggestions mentioned by our worthy reviewers and in accordance of reviewer’s comments, we have revised the manuscript. 

In general, all the recommendations and suggestions have been addressed and incorporated in the manuscript which include following.

Regards

Mariusz Mojzych

Reviewer 2 Report

Referee comments for Manuscript ID: molecules-1956420

The manuscript entitled “Mitsunobu Reaction: A Powerful Tool for the Synthesis of 2 Natural Products: A Review”

Recommendation: Accept in present form.

The authors have reported a review on the Mitsunobu reaction in organic chemistry and synthetic applications. In addition, the authors summarize the role of the Mitsunobu reaction in the synthesis of natural products. Finally, the authors focus on the contribution of Mitsunobu reaction towards the total synthesis of specific natural products highlighting their biological potential during recent years.

This review has many nice aspects, and the author describes in detail and consistently the important of Mitsunobu reaction and highlighting their biological potential during recent years.

I think it should be accepted in present form apart of minor point regarding Scheme 2. “Synthesis of deoxyuzurimine alkaloid” should be fitted to the size of the paper.

Best regards,

Dr. Abed Saady

Author Response

(The authors gave the same response as above.)

Reviewer 3 Report

Summary of the key contribution of the paper:

Mitsunobu Reaction: A Powerful Tool for the Synthesis of Natural Products: A Review gives an overview and a discussion of The Mitsunobu reaction plays a vital part in organic chemistry due to its wide synthetic applications. This review article will focus on the contribution of Mitsunobu reaction towards the total synthesis of natural products highlighting their biological potential during recent years.

Highlights:

·        Review clearly articulates the use a discussion of Mitsunobu reaction has been in use for past few decades in a wide variety of medical applications

·        Figures and tables are well referenced and clear

·        Throughout this review, the significance of Mitsunobu reaction in the total synthesis of natural products from recent years has been mentioned and explained application and research is up to date

Author Response

Dear Sir,

Thank you very much for peer reviewing our manuscript and we appreciate your complimentary recommendations .

Regards

Mariusz Mojzych